# Pleiotropic Effects of Functional *MUC1* Variants on Cardiometabolic, Renal, and Hematological Traits in the Taiwanese Population

**DOI:** 10.3390/ijms221910641

**Published:** 2021-09-30

**Authors:** Ming-Sheng Teng, Semon Wu, Lung-An Hsu, Hsin-Hua Chou, Yu-Lin Ko

**Affiliations:** 1Department of Research, Taipei Tzu Chi Hospital, Buddhist Tzu Chi Medical Foundation, New Taipei City 23142, Taiwan; vincent@tzuchi.com.tw; 2Department of Life Science, Chinese Culture University, Taipei 11114, Taiwan; semonwu@yahoo.com.tw; 3The First Cardiovascular Division, Department of Internal Medicine, Chang Gung Memorial Hospital and Chang Gung University College of Medicine, Taoyuan 33305, Taiwan; hsula@adm.cgmh.org.tw; 4The Division of Cardiology, Department of Internal Medicine and Cardiovascular Center, Taipei Tzu Chi Hospital, Buddhist Tzu Chi Medical Foundation, New Taipei City 23142, Taiwan; chouhhtw@gmail.com; 5School of Medicine, Tzu Chi University, Hualien 97004, Taiwan

**Keywords:** *MUC1* gene, polymorphism, pleiotropic effect, DNA methylation, regional plot association study

## Abstract

*MUC1* is a transmembrane mucin involved in carcinogenesis and cell signaling. Functional *MUC1* variants are associated with multiple metabolic and biochemical traits. This study investigated the association of functional *MUC1* variants with *MUC1* DNA methylation and various metabolic, biochemical, and hematological parameters. In total, 80,728 participants from the Taiwan Biobank were enrolled for association analysis using functional *MUC1* variants and a nearby gene regional plot association study. A subgroup of 1686 participants was recruited for *MUC1* DNA methylation analysis. After Bonferroni correction, we found that two *MUC1* variants, rs4072037 and rs12411216, were significantly associated with waist circumference, systolic blood pressure, hemoglobin A1C, renal functional parameters (blood urea nitrogen, serum creatinine levels, and estimated glomerular filtration rate), albuminuria, hematocrit, hemoglobin, red blood cell count, serum uric acid level, and gout risk, with both favorable and unfavorable effects. Causal inference analysis revealed that the association between the variants and gout was partially dependent on the serum uric acid level. Both gene variants showed genome-wide significant associations with *MUC1* gene-body methylation. Regional plot association analysis further revealed lead single-nucleotide polymorphisms situated at the nearby *TRIM46*–*MUC1*–*THBS3*–*MTX1* gene region for the studied phenotypes. In conclusion, our data demonstrated the pleiotropic effects of *MUC1* variants with novel associations for gout, red blood cell parameters, and *MUC1* DNA methylation. These results provide further evidence in understanding the critical role of *TRIM46*–*MUC1*–*THBS3*–*MTX1* gene region variants in the pathogenesis of cardiometabolic, renal, and hematological disorders.

## 1. Introduction

Mucin family members are classified into two types on the basis of their localization: secreted or membranous [1,2,3,4]. Mucin 1 (*MUC1*), also named polymorphic epithelial mucin and encoded as *MUC1* in humans, is a single-pass type I transmembrane phosphoprotein with substantial O-linked glycosylation. In terms of its functions, *MUC1* was initially assumed to only protect from the external environment and lubricate the epithelium, whereas *MUC1* is currently considered to play a crucial role as a multifunctional protein in cell signal transduction and cell–cell interaction [1,3]. *MUC1* is normally expressed in the glandular or luminal epithelial cells of the mammary gland, esophagus, stomach, duodenum, pancreas, uterus, prostate, and lungs, and to a lesser extent, in hematopoietic cells [5,6,7]. Furthermore, *MUC1* is overexpressed in breast, ovarian, lung, pancreatic, and prostate cancers and is a marker of poor prognosis in gastric cancer [8,9]. A recent proteomic analysis of urine revealed that the urinary excretion of five proteins including *MUC1* is associated with the risk of renal impairment in the general population. In addition, this study indicated that the diagnostic value of urinary *MUC1* in predicting a decline in the estimated glomerular filtration rate (eGFR) was higher than that of microalbuminuria [10].

*MUC1*, located on chromosome 1q22, is a causative gene for autosomal dominant tubulointerstitial kidney disease (ADTKD), which is a hereditary disease characterized by progressive tubulointerstitial nephropathy that ultimately leads to end-stage renal disease (ESRD) [11,12,13]. Several genome-wide association studies (GWASs) have revealed a significant association of *MUC1* polymorphisms with the risk of gastric cancer [14,15,16]. In addition, recent GWASs reported associations of two *MUC1* single-nucleotide polymorphisms (SNPs), namely the functional variants rs4072037 and rs12411216, with serum magnesium and uric acid levels, blood pressure, impaired renal function, and albuminuria [17,18,19,20,21]. The Taiwan Biobank (TWB) conducted a large-scale population-based cohort study by recruiting volunteers aged between 30 and 70 years with no history of cancer [22,23]. In the present study, we hypothesized that both genetic and epigenetic effects of the *MUC1* gene may play a crucial role in the pathogenesis of cardiometabolic, renal, and hematological disorders. We investigated the association of functional *MUC1* variants with various metabolic, biochemical, and hematological parameters in more than 80,000 TWB participants and with *MUC1* DNA methylation in 1686 TWB participants. In addition, we examined whether causal inference occurred between the associated phenotypes, identified variants that played more crucial roles in determining the association with various phenotypes, and evaluated the role of nearby gene loci variants by performing a regional plot association study.

## 2. Results

### 2.1. Characteristics of Clinical, Biochemical, and Hematological Traits

Table 1 lists the demographic data, clinical and biochemical data, lipid profiles, and hematological traits of TWB participants stratified by sex. Compared with women, men had a significantly higher BMI, waist circumference, waist to hip ratio, systolic, diastolic, and mean blood pressure, fasting plasma glucose, serum triglyceride, uric acid, creatinine, BUN, AST, ALT, γ-GT, albumin, total bilirubin levels, hematocrit, red blood cell count, and hemoglobin (all *p* < 0.0001). By contrast, HDL-cholesterol, total cholesterol levels, eGFR, and platelet counts (all *p* < 0.0001) were lower in men than in women.

### 2.2. Association of MUC1 Genotypes with Clinical, Metabolic, and Biochemical Phenotypes and Hematological Parameters

A total of over 80,000 volunteers participated in the genotype–phenotype association analysis. We analyzed the association of *MUC1* rs12411216 and rs4072037 genotypes with various clinical phenotypes and laboratory parameters. By using an additive model, after adjustment for age, sex, BMI, and smoking status and after Bonferroni correction, we observed a significant association of both *MUC1* variants with waist circumference, systolic blood pressure, HbA1C, renal functional parameters (BUN, serum creatinine levels, and eGFR), albuminuria, red blood cell parameters (hematocrit, hemoglobin, and red blood cell count), serum ALT and uric acid levels, and gout (Table 2). A near complete linkage between the rs12411216 and rs4072037 polymorphisms was found with an r^2^ of 0.9832.

### 2.3. Association of MUC1 Genotypes with Risk Factors for Atherosclerosis

We examined the association of *MUC1* rs12411216 and rs4072037 genotypes with risk factors for atherosclerosis. The results revealed that gout and microalbuminuria were significantly associated with *MUC1* gene variants (Figure 1 and Appendix A).

### 2.4. Subgroup Analysis of the Association of MUC1 Genotypes with Studied Parameters and Atherosclerotic Risk Factors

We investigated whether sex affects the association between the two variants and the studied phenotypes (Appendix A). The results of the *t* test revealed that sex affected only the association between the *MUC1* variants and albuminuria (*t* = −2.6977 and *p* < 0.01 for the rs12411216 genotype and *t* = −2.7698 and *p* < 0.01 for the rs4072037 genotype). Both the variants showed a more significant association with albuminuria in men.

### 2.5. Mediational Analysis: MUC1 Genotypes and Various Phenotypes

The two-stage least square instrumental variable (IV) analysis was performed to examine the causality between the studied phenotypes such as the eGFR, hematocrit, uric acid level, gout, and albuminuria with rs4072037 and rs12411216 genotypes used as IVs (Appendix A). Our data revealed an association between the *MUC1* variants and the studied phenotypes, and this association remained highly significant after adjustment for other phenotypes, suggesting no evidence of a causal relationship between these study phenotypes. The only exception was that the association between each variant and gout disappeared after adjustment for the serum uric acid level (*p* value decreased from 2.94 × 10^−8^ to 0.0004 and from 3.66 × 10^−8^ to 0.0006 for rs4072037 and rs12411216 genotypes, respectively). These results suggested that the association between both variants and gout would be at least partially mediated by the serum uric acid level.

### 2.6. Association between the Functional Polymorphisms of the MUC1 Gene and the Nearby MUC1 DNA Methylation Status

We further examined whether the two *MUC1* functional variants are associated with the DNA methylation status of the *MUC1* gene. We found no significant association between the two variants and 12 *MUC1* promoter methylation sites (Appendix A). By contrast, genome-wide significant associations were observed between both variants and four *MUC1* gene-body DNA methylation sites, namely cg06339768, cg19011149, cg15646096, and cg00126087, with a stronger association noted for the rs4072037 genotype, as shown in Figure 2 (*p* = 1.01 × 10^−^^28^, 4.23 × 10^−^^66^, 1.34 × 10^−^^121^, and 5.84 × 10^−^^27^, respectively, after adjustment for age, sex, BMI, and smoking with Bonferroni correction).

The C allele of rs12411216 and the A allele of rs4072037 were significantly associated with the hypermethylated status of the *MUC1* gene body (Figure 3A–H and Appendix A).

Furthermore, we analyzed the association between cg15646096 and the studied clinical and laboratory parameters and found that the methylation status of cg15646096 significantly decreased with age (Appendix A).

### 2.7. Regional Plot Association Studies for Genetic Variants at Positions 155 to 155.5 Mb on Chromosome 1q22 and the Study Phenotypes

We performed a regional plot analysis to examine the association between genetic variants from position 155.0 to 155.5 Mb on chromosome 1q22 and the studied phenotypes. Genome-wide significant associations for the lead SNPs were noted for uric acid level (rs2070803, *p* = 3.63 × 10^−33^), gout (rs12411216, *p* = 2.94 × 10^−8^), BUN (rs4971101, *p* = 2.78 × 10^−35^), eGFR (rs2070803, *p* = 3.06 × 10^−19^), albuminuria (rs423144, *p* = 2.21 × 10^−10^), hematocrit (rs4971093, *p* = 2.1 × 10^−15^), hemoglobin (rs76872124, *p* = 2.27 × 10^−14^), and red blood cell count (rs76872124, *p* = 1.18 × 10^−9^). Our data revealed that the lead SNP for each phenotype was situated at the *TRIM46*–*MUC1*–*THBS3*–*MTX1* gene region (Figure 4A, Appendix A, and Table 3). After conditional analysis, we observed a genome-wide significant association of only *TRIM5* rs76872124 with BUN after adjustment for the lead SNP rs4971101 (Appendix A). Most of the lead SNPs showed strong LD with rs4072037 and rs12411216 (with r^2^ > 0.87), whereas the lead SNPs for red blood cell parameters, such as rs4971093 and rs76872124, exhibited a moderate LD with rs4072037 (r^2^ = 0.705 and r^2^ = 0.544, respectively). The results are summarized in Table 3 and Figure 4B.

## 3. Discussion

To the best of our knowledge, this is the first study to investigate the association of *MUC1* functional SNPs with multiple clinical and laboratory parameters by using both candidate gene and regional plot association approaches. We observed that both *MUC1* variants were associated with renal functional parameters (BUN, serum creatinine level, and eGFR), albuminuria, and serum uric acid levels; this finding is in accordance with that of previous studies. Furthermore, we identified novel genome-wide significant associations between *MUC1* functional SNPs and gout, red blood cell parameters (hematocrit, hemoglobin, and red blood cell counts), and *MUC1* gene-body DNA methylation. Our results suggested that both SNPs play a critical role in determining different phenotypes. We also provide the first evidence of the possibility that the previously reported association between rs4072037 and the total or alternative spliced form of gene expression may be secondary to *MUC1* gene-body methylation status. The findings of causal inference analysis showed that the association of both variants with gout was partially dependent on the serum uric acid level. No cause–effect relationship was found in other phenotypes, indicating mostly pleiotropic effects. The results of regional plot association analysis revealed lead SNPs at the nearby *TRIM46*–*MUC1*–*THBS3*–*MTX1* gene region for the studied phenotypes, providing further evidence for the critical role of *MUC1* as a candidate gene locus for cardiometabolic, renal, and hematological disorders.

### 3.1. Association between MUC1 Polymorphisms and Renal Function and Albuminuria

In a GWAS analyzing molecular insights into chronic kidney disease-defining traits, Xu et al. [20] reported the causal effects of three genes (*NAT8B*, *CASP9*, and *MUC1*) on the eGFR, identified rs4072037 as a common alternative splice variant in *MUC1*, and observed increased renal expression of a specific *MUC1* mRNA isoform as a plausible molecular mechanism underlying the GWAS association signal. Moreover, according to combined annotation-dependent depletion scores, rs4072037 and rs12411216 belonged to the top 6% and 10%, respectively, of most functionally significant SNPs in the human genome [20]. Of these, rs12411216 maps onto the CpG island within the promoter region for *MUC1*, whereas rs4072037 operates as an alternative splice site acceptor. Our data showed a genome-wide significant association of *MUC1* variants with BUN and eGFR in TWB participants, thus supporting the critical role of *MUC1* in renal functional impairment.

Higher albuminuria is associated with adverse clinical outcomes such as ESRD, cardiovascular disease, and mortality [24]. Teumer et al. [21] performed a GWAS to identify genetic loci associated with albuminuria in diabetes by analyzing 7877 participants of European descent and found *MUC1* as one of the candidate loci. By performing trans-ethnic (*n* = 564,257) and European-ancestry-specific meta-analyses of the GWASs of the urine albumin-to-creatinine ratio (UACR), including ancestry- and diabetes-specific analyses, Teumer et al. [25] identified 68 UACR-associated loci and showed a genetic correlation among proteinuria, hyperlipidemia, gout, and hypertension. Furthermore, fine-mapping and trans-omics analyses with gene expression showed *MUC1* as a gene potentially operating through differential expression in the kidney. With only the urine albumin level available for analysis, our data also revealed a genome-wide significant association between albuminuria and both *MUC1* variants. Moreover, a lower frequency of microalbuminuria was found in the C allele of rs12411216 and the A allele of rs4072037, supporting *MUC1* as a candidate gene for albuminuria in both European and Asian populations.

### 3.2. MUC1 Polymorphisms and Serum Uric Acid Levels and Gout

Two GWASs including 109,029 and 121,745 Japanese individuals have shown rs4072037 as the lead SNP for the uric acid level in the *MUCI* gene region [17,26]. In addition, our results showed a strong association of rs4072037 and rs12411216 with both the serum uric acid level and gout. Furthermore, Sandoval-Plata et al. [27] indicated that the nearby *MTX1* SNP rs760077 showed a significant genome-wide association with gout compared with normal uric acid level controls in UK Biobank participants. All these findings suggest the importance of *MUC1* in the development of hyperuricemia and gout.

### 3.3. MUC1 Polymorphisms and Blood Pressure Status and Waist Circumference

Our results demonstrated a borderline significant association of the *MUC1* rs4072037 genotype with blood pressure status. Takeuchi et al. [19] analyzed 289,038 individuals of East Asian and European descent and found a significant genome-wide association between *MUC1* rs4072037 and diastolic blood pressure. Moreover, ADTKD-MUC1 with the *MUC1* mutation was found to be associated with ESRD and hypertension. For waist circumference, the lead SNP determined in our study was *MTX1* rs760077 (*p* = 0.0001; Table 3), which showed a near complete linkage with rs4072037 (r^2^ = 0.965). Recently, a cross-trait GWAS mapped the *THBS3* SNP rs72704117 to waist circumference using participants from the UK Biobank [28]. Their results indicated that *MUC1* and its nearby gene region may play a critical role in determining blood pressure status and central obesity.

### 3.4. Association between MUC1 Polymorphisms and Red Blood Cell Parameters

*MUC1* can be expressed in nonepithelial cells including hematopoietic cells [7,29]. In bone marrow differentiating cells, *MUC1* was strongly and selectively expressed during erythropoiesis but was absent in circulating erythrocytes. During erythropoiesis, *MUC1* expression was transcriptionally regulated through phosphorylation. The sialylated *MUC1* glycoforms selectively expressed on erythroid cells can act as a ligand for CD169, a macrophage-restricted adhesion molecule that is vital for erythropoiesis. These results suggested that *MUC1* acts as a cross-talk molecule between erythroblasts and surrounding cells during erythropoiesis [7]. In addition, *MUC1* is overexpressed and aberrantly glycosylated in hematological malignancies [1]. Our data revealed that both *MUC1* variants were significantly associated with red blood cell parameters including hematocrit, hemoglobin, and red blood cell counts. A recent study examining variations in human blood cell traits reported *MTX1* rs760077 as a candidate variant associated with all the three red cell parameters [30].

### 3.5. Previous Functional Studies for MUC1 rs4072037 and rs12411216 Polymorphisms

Both *MUC1* rs4072037 and rs12411216 polymorphisms have been shown to be functional. *MUC1* rs4072037 located in the 5’ side of exon 2 operates as an alternative splice site acceptor. By conducting the RNA-ligase-mediated rapid amplification of the 5′ complementary DNA end procedure, the rs4072037 variant was found completely associated with two major *MUC1* transcripts in the gastric epithelium: variants 2 and 3 [15]. The allele A at rs4072037 causes a 9-amino acid deletion in exon 2 and consequently modifies both the signal peptide and N-terminal amino acid of the mature protein by changing the signal peptide cleavage site [31]. The effect of rs4072037 on the *MUC1* mRNA alternative splicing isoform, the second most abundant isoform in the kidney, was examined by performing transcriptome analysis [20]. Carriers of one and two copies of the allele A of this splice variant exhibited the intermediate and highest expression levels of the alternatively spliced *MUC1* isoform and the total renal expression of *MUC1* gene when compared with the reference genotype, suggesting that the rs4072037-driven allelic effect on the expression of a specific *MUC1* mRNA isoform can be the key biological mechanism underlying the genetic association observed in previous GWASs. In addition, the association of the rs4072037 genotypes with various phenotypes may be due to a strong LD with other functional SNPs. *EFNA1* rs12904, an miR-200c binding site SNP that shows a strong LD with rs4072037 and is located in the 3’-untranslated region of the *EFNA1* gene, can modulate *EFNA1* expression and is associated with gastric cancer susceptibility [32]. The *MUC1* rs12411216 polymorphism is located in the CpG island within the promoter region for *MUC1* and is in near complete LD with rs4072037. Jiang et al. [33] revealed that the rs12411216 genotype was associated with mild cognitive impairment and glucocerebrosidase (GBA) expression in Parkinson’s disease. By performing an electrophoretic mobility shift assay, the authors reported that the rs12411216 polymorphism could affect the binding efficiency of the transcription factor E2F4 and regulate GBA expression. Low GBA expression promoted the prion-like spread of α-Syn interpolymer complexes, facilitated the pathogenesis of Parkinson’s disease, and increased cognitive damage [34,35]. These results characterized rs12411216 as a pathogenic variant of GBA in patients with Parkinson’s disease with mild cognitive impairment.

### 3.6. Association between Functional MUC1 Variants and MUC1 Gene Methylation

DNA methylation is a crucial epigenetic marker, with the cytosine of CpG dinucleotide being the major methylation site. Hypermethylated CpG islands in promoters may silence the transcription of genes [36]. In gene-body methylation, exons were more highly methylated than introns, and sharp transitions of methylation occurred at exon–intron boundaries. Gene-body methylation is highly conserved across eukaryotes; in contrast to promoter methylation, gene-body methylation is positively associated with transcription but not linked to gene silencing [37,38]. DNA methylation in gene bodies may facilitate transcription elongation and/or co-transcriptional splicing and may repress intragenic cryptic promoters [39]. CCCTC binding factor, a DNA-binding protein that is bound to exon 5 of the *CD45* gene, may be inhibited by *CD45* gene-body DNA methylation, which results in local RNA polymerase II pause, reducing the transcription elongation rate, and favoring the co-transcriptional spliceosome assembly to promote weak upstream exon inclusion and alterative splicing [40]. Intragenic DNA methylation can modulate alternative splicing by recruiting methyl-CpG-binding protein to promote exon recognition [41]. Xu et al. [20] examined whether rs12411216 affects *MUC1* promoter methylation and found that none of the CpG sites within the CpG island overlaying the *MUC1* promoter region showed an association with the rs12411216 genotype. Our data also revealed no significant association between two functional *MUC1* variants and the methylation status of 12 *MUC1* promoter CpG sites. By contrast, we observed genome-wide significant associations between both rs4072037 and rs12411216 genotypes and four DNA methylation sites at the gene body of *MUC1*. Xu et al. [20] showed that the risk allele A at rs4072037 may increase the expression levels of the alternatively spliced *MUC1* isoform. In our study, this risk allele was significantly associated with a highly methylation status at the *MCU1* gene body. Thus, the possibility that the association between rs4072037 and the total or alternative spliced form of gene expression may be secondary to *MUC1* gene-body methylation status should be highly considered. However, epigenetic effects are tissue-specific, and additional studies may be necessary to confirm the conclusion in different tissues and disease statuses.

### 3.7. Role of Nearby Gene Region Variants Determined by Performing Regional Plot Association Studies

By performing regional plot association studies, we tested whether the association between *MUC1* functional variants and various phenotypes may be due to LD with adjacent gene variants. When the data of the lead SNPs and two functional *MUC1* variants were combined in the regional plot analysis, we observed that a major association with the study phenotypes was focused on variants at the nearby *TRIM46*–*MUC1*–*THBS3*–*MTX1* gene region with at least moderate LD between SNPs (Figure 4A,B). The associations of the A allele of the rs4072037 genotype and the C allele of the rs12411216 genotype with lower systolic blood pressure, uric acid levels, and gout risk and a higher eGFR and hematocrit may be beneficial to participants, whereas the association with higher waist circumference, HbA1C, albuminuria, gastric cancer risk, and lower bone mineral density may be unfavorable to participants.

### 3.8. Limitations

Because only urine albumin but not urine creatinine was measured, the UACR could not be determined in our study. However, the association between *MUC1* variants and albuminuria has been confirmed by previous studies [21,25].

## 4. Participants and Methods

### 4.1. TWB Participants

The study cohort for the regional plot association study consisted of participants with samples in the TWB; they were recruited from centers across Taiwan between 2008 and 2020. A total of 107,494 participants with no history of cancer and with genotyping from Axiom Genome-Wide CHB 1 or 2 Array were recruited, and 26,766 participants were excluded from the analysis based on the following criteria: no imputation data (12,289 participants), quality control (QC) for the GWAS (10,956 participants), fasting for <6 h (2862 participants), failure to genotype the rs4072037 or rs12411216 polymorphism (371 participants), and the absence of any study phenotype (288 participants). During QC for the GWAS, all of the participants excluded were due to relative pairs of 2nd degree relatives or closer with identity by descent (IBD) > 0.187. While examining the serum uric acid level, blood pressure status, and lipid and glucose metabolism parameters, we excluded participants with a history of gout, hypertension, hyperlipidemia, and diabetes mellitus, respectively. Figure 5 depicts the flowchart of participant enrollment. The definitions of hypertension, diabetes mellitus, obesity, hyperlipidemia, and current smoking are provided in Appendix A. Ethical approval was received from the Research Ethics Committee of Taipei Tzu Chi Hospital, Buddhist Tzu Chi Medical Foundation (approval number: 08-XD-005), and Ethics and Governance Council of the Taiwan Biobank (approval number: TWBR10908-01). Each participant signed an informed consent form before participation in the study.

### 4.2. Clinical Phenotypes and Laboratory Examinations

We examined the following clinical phenotypes: body height; body weight; waist circumference; waist–hip ratio; body mass index (BMI); and systolic, mean, and diastolic blood pressure. In addition, we collected the following biochemistry data: lipid profiles such as total, high-density lipoprotein (HDL), and low-density lipoprotein (LDL) cholesterol and triglyceride levels, glucose metabolism parameters such as fasting plasma glucose and hemoglobin A1C (HbA1C) levels, and liver and renal functional test-related parameters such as blood urea nitrogen (BUN), serum creatinine, uric acid, aspartate aminotransferase (AST), alanine aminotransferase (ALT), γ-glutamyl transferase (γ-GT), albumin, and total bilirubin levels. The BMI and eGFR were calculated as reported previously [42]. The hematological parameters analyzed included white and red blood cell counts, platelet counts, and hematocrit and hemoglobin levels. Because of the absence of the urine creatinine level, only the spot urine albumin level was used to evaluate albuminuria. Microalbuminuria was defined as a urine albumin level of ≥30 mg/L.

### 4.3. Selection of Functional MUC1 Variants and Genotyping

For TWB participants, DNA was isolated from blood samples by using a PerkinElmer Chemagic 360 instrument (PerkinElmer, Waltham, MA, USA) following the manufacturer’s instructions. SNP genotyping was conducted using custom TWB chips and performed on the Axiom genome-wide array plate system (Affymetrix, Santa Clara, CA, USA). Two previously reported functional *MUC1* variants, rs4072037 and rs12411216, were used for the analysis [15,31] (Appendix A).

### 4.4. DNA Methylation Analysis

DNA methylation was assessed using sodium-bisulfite-treated DNA from whole blood by using the Infinium MethylationEPIC BeadChipEPIC array (Illumina Inc. San Diego, CA, USA). Four quality control measures were used: correction for dye bias across batches by normalization, removal of background signals, elimination of outliers by the median absolute deviation method, and elimination of probes with poor detection (*p* > 0.05) and those whose bead counts were <3. A total of 1686 participants were enrolled for the analysis of *MUC1* DNA methylation.

### 4.5. Regional Plot Association Analysis

To determine the lead SNP variants around the *MUC1* region for various studied phenotypes, we first performed a quality check (QC) for GWAS. The Axiom genome-wide CHB 1 and 2 array plates (Affymetrix, Inc., Santa Clara, CA, USA), each with 27,720 and 67,485 participants, and comprising 611,656 and 640,160 SNPs, respectively, underwent the imputation analysis. By using the East Asian population from the 1000 Genome Project Phase 3 study as the reference panel, genome-wide genotype imputation was performed using SHAPEIT and IMPUTE2. After imputation, QC was performed by filtering SNPs with an IMPUTE2 imputation quality score of >0.3. Indels were removed using VCFtools. All samples enrolled for the analysis had a call rate of ≥97% with IBD ≥ 0.187. For SNP QC, an SNP call rate of <3%, a minor allele frequency of <0.01, and the violation of the Hardy–Weinberg equilibrium (*p* < 10^−6^) were the criteria for exclusion from subsequent analyses. Finally, after QC, 84,249 participants and 3639,842 SNPs remained in the whole genome and 236 SNPs at positions between 155.0 and 155.5 Mb on chromosome 1q22 were enrolled for the regional plot association analysis.

### 4.6. Statistical Analysis

Continuous variables are expressed as the mean ± standard deviation. When the distribution was strongly skewed, median and interquartile ranges are presented that were examined using a two-sample t test or analysis of variance. Differences in categorical data distribution were examined using a chi-squared test or chi-squared test for trend. Before analysis, all study parameters were logarithmically transformed to adhere to a normality assumption. We assumed the genetic effect to be additive after adjustment for age, sex, BMI, and current smoking status, and a generalized linear model was used to analyze studied phenotypes in relation to the predictors of investigated genotypes and confounders. Regional plot association studies were performed using the analysis software package PLINK (version 1.07, Shaun Purcell, Cambridge, MA, USA, http://pngu.mgh.harvard.edu/purcell/plink, accessed on 14 August 2021). Genome-wide significance was defined as a *p* value of <5 × 10^−8^. The LDmatrix software (https://analysistools.nci.nih.gov/LDlink/?tab=ldmatrix, accessed on 19 April 2021) was used for the analysis of linkage disequilibrium (LD). All statistical analyses were performed using SPSS (version 22; SPSS, Chicago, IL, USA). A two-sided *p* value of <0.05 was considered to be statistically significant.

## 5. Conclusions

Our data confirmed the presence of pleiotropy at the *TRIM46–MUC–THBS3–MTX1 gene region*, especially at *MUC1* gene functional variants, with multiple phenotypes being associated with candidate variants. Our data linked rs4072037 to *MUC1* gene-body methylation, which may affect alternative splicing and increase the expression of *MUC1* alternative splicing isoforms. The bidirectional effect of functional *MUC1* gene variants on health should be examined in future studies. These results can provide insights into the critical role of *MUC1* in the pathogenesis of cardiometabolic, renal, and hematological disorders.

## Figures and Tables

**Figure 1 ijms-22-10641-f001:**
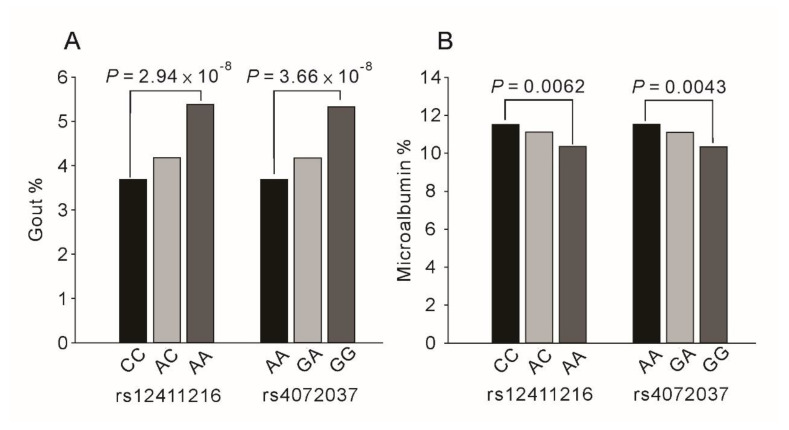
Association between *MUC1* rs12411216 and rs4072037 genotypes for gout (**A**) and microalbuminuria (**B**). *p* values were adjusted for age, sex, body mass index, and current smoking. Significantly higher frequencies of gout and lower frequencies of microalbuminuria were noted for A allele of rs407203 and C allele of rs12411216.

**Figure 2 ijms-22-10641-f002:**
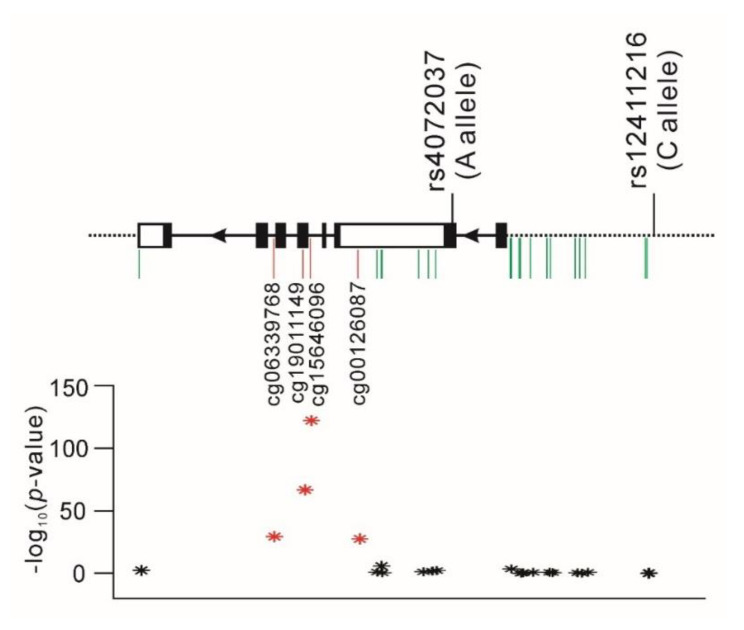
Genomic structure of functional *MUC1* variants and their association with *MUC1* gene DNA methylation. Regional plot analysis of association between the A allele of rs4072037 and the C allele of rs12411216 and DNA methylation. *: *p* < 5.00 × 10^−8^, *: *p* ≥ 5.00 × 10^−8^.

**Figure 3 ijms-22-10641-f003:**
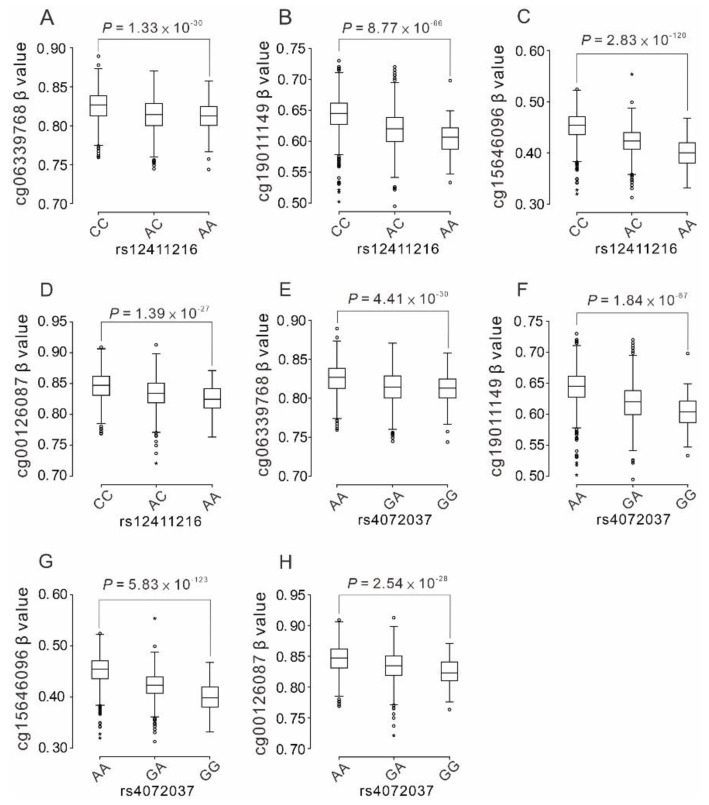
*MUC1* gene-body DNA methylation levels according to *MUC1* variants. The box-and-whisker plots are based on 1686 Taiwan Biobank participants with adjustment of age, sex, body mass index, and current smoking. The associations of cg06339768, cg19011149, cg15646096, and cg00126087 with rs12411216 genotypes (**A**–**D**) and with rs4072037 genotypes (**E**–**H**) are shown. o: β value < mean ± 3 standard deviation (SD), *: β value ≥ mean ± 3 SD.

**Figure 4 ijms-22-10641-f004:**
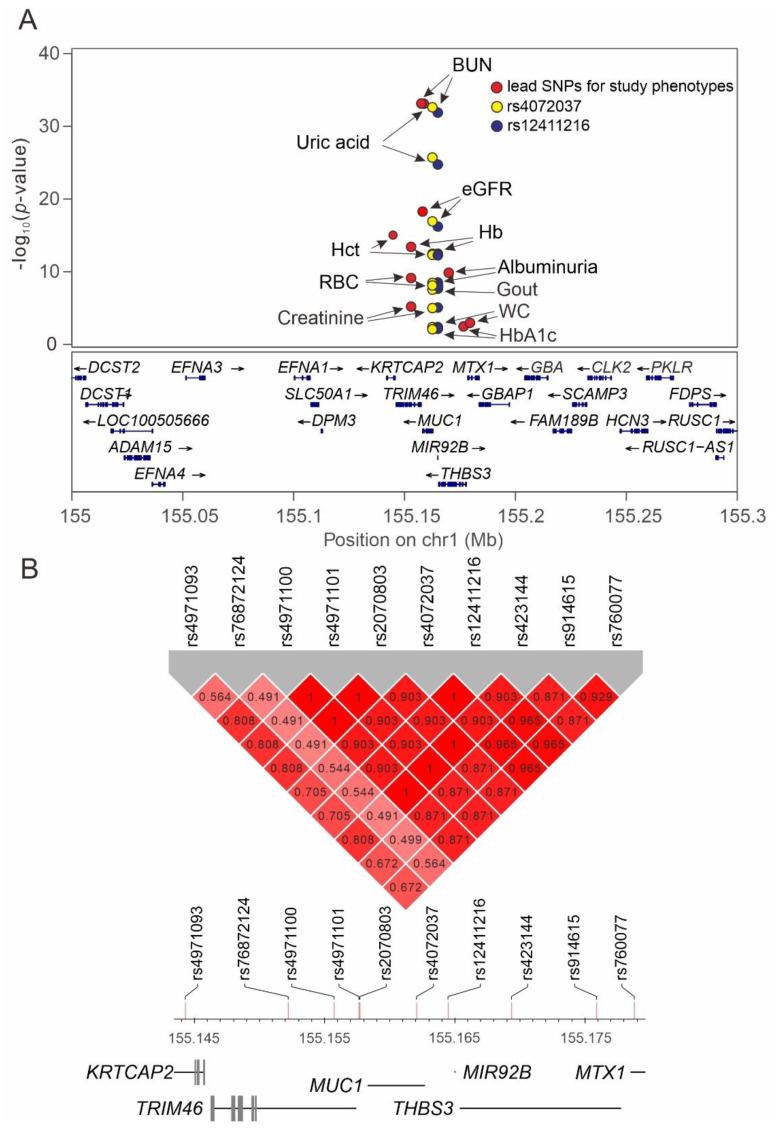
Combination of regional plot association studies between genetic variants at positions 155.0 to 155.3 mega-base (Mb) on chromosome 1q22 and multiple phenotypes. (**A**) For each phenotype, single-nucleotide polymorphism (SNP) rs12411216 is shown in purple, SNP rs4072037 is shown in yellow, and other lead SNPs are shown in red, except for gout in which the lead SNP was rs12411216. The lead SNP for each phenotype is indicated by an arrow. (**B**) Linkage disequilibrium maps for the *MUC1* functional variants and lead SNPs in the *TRIM46–MUC1–THBS3–MTX1* gene region. BUN: blood urea nitrogen, eGFR: estimated glomerular filtration rate, Hb: hemoglobin, Hct: hematocrit, RBC: red blood cell, WC: waist circumference.

**Figure 5 ijms-22-10641-f005:**
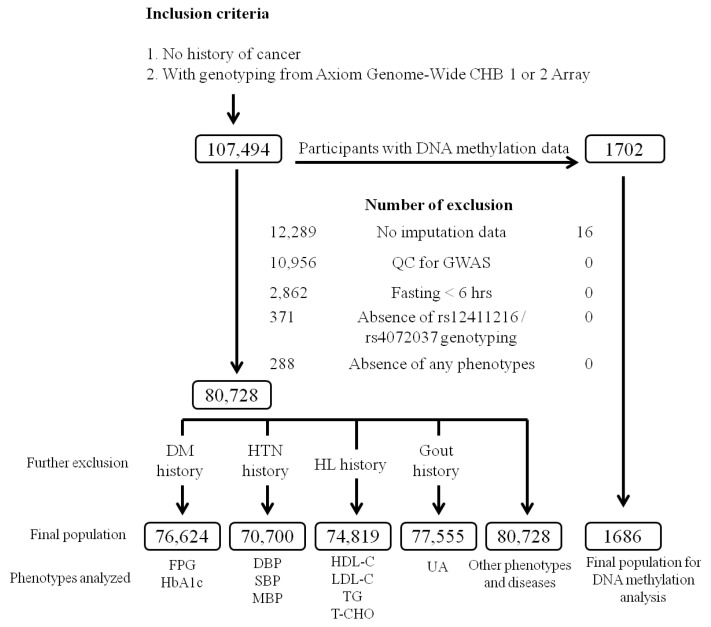
Study inclusion and exclusion criteria flowchart. This flowchart presents the inclusion and exclusion criteria used to screen for Taiwan Biobank (TWB) participants. GWAS: genome-wide association study, QC: quality control, HL: hyperlipidemia, HTN: hypertension, DM: diabetes mellitus, FPG: fasting plasma glucose, HbA1C: hemoglobin A1C, SBP: systolic blood pressure, DBP: diastolic blood pressure, MBP: mean blood pressure, LDL-C: low-density lipoprotein cholesterol, HDL-C: high-density lipoprotein cholesterol, TG: triglyceride, T-CHO: total cholesterol, UA: uric acid. Other phenotypes include: age, body mass index (BMI), waist circumference, waist–hip ratio, aspartate aminotransferase (AST), alanine aminotransferase (ALT), γ-Glutamyl transferase (γ-GT), serum creatinine level, estimated glomerular filtration rate (eGFR), serum albumin, total bilirubin, hemoglobin, hematocrit, red blood cell, leukocyte, and platelet counts, blood urea nitrogen (BUN), albuminuria, microalbuminuria, current smoking status, metabolic syndrome.

**Table 1 ijms-22-10641-t001:** Baseline characteristics of study subjects according to sex.

Clinical and Laboratory Parameters		Male	Female	*p* Value
	Number	29,266	51,462	
Anthropology	Age (years)	51.0 (41.0–60.0)	51.0 (41.0–59.0)	9.52 × 10^−10^
	Waist circumference (cm)	87.5 (82.0–93.5)	80.0 (74.0–86.5)	<10^−307^
	Waist–hip ratio	0.90 (0.86–0.94)	0.84 (0.80–0.89)	<10^−307^
	Body mass index (kg/m^2^)	25.0 (23.0–27.3)	23.0 (21.0–25.5)	<10^−307^
Blood pressure	Systolic BP * (mmHg)	121.0 (112.0–131.5)	111.0 (102.0–123.0)	<10^−307^
	Diastolic BP * (mmHg)	76.5 (70.0–83.0)	69.0 (62.7–76.0)	<10^−307^
	Mean BP * (mmHg)	91.2 (84.3–98.7)	83.2 (76.3–91.3)	<10^−307^
Lipid profiles	Total cholesterol **** (mg/dL)	190.0 (168.0–213.0)	194.0 (172.0–218.0)	3.77 × 10^−131^
	HDL-cholesterol **** (mg/dL)	47.0 (40.0–54.0)	57.0 (49.0–66.0)	<10^−307^
	LDL-cholesterol **** (mg/dL)	121.0 (101.0–142.0)	118.0 (98.0–140.0)	0.1872
	Triglyceride **** (mg/dL)	107.0 (74.0–156.0)	83.0 (59.0–119.0)	2.36 × 10^−223^
Glucose metabolism	Fasting plasma glucose ** (mg/dL)	94.0 (89.0–99.0)	90.0 (86.0–95.0)	8.29 × 10^−103^
	HbA1C ** (%)	5.6 (5.4–5.9)	5.6 (5.4–5.8)	0.8662
Uric acid	Uric acid *** (mg/dL)	6.3 (5.5–7.1)	4.8 (4.1–5.5)	<10^−307^
Renal function	Creatinine (mg/dL)	0.88 (0.79–0.98)	0.60 (0.54–0.67)	<10^−307^
	eGFR (mL/min/1.73 m^2^)	92.6 (81.3–105.0)	106.2 (92.4–122.2)	<10^−307^
	BUN (mg/dL)	13.6 (11.5–16.0)	12.2 (10.1–14.6)	<10^−307^
	Albuminuria (mg/L)	8.4 (5.3–15.0)	9.0 (5.5–15.4)	3.53 × 10^−29^
Liver function	AST (U/L)	24.0 (21.0–29.0)	22.0 (19.0–26.0)	5.09 × 10^−119^
	ALT (U/L)	23.0 (17.0–33.0)	17.0 (13.0–23.0)	<10^−307^
	γ-GT (U/L)	22.0 (16.0–34.0)	15.0 (11.0–21.0)	2.26 × 10^−254^
	Serum albumin (g/dL)	4.6 (4.4–4.7)	4.5 (4.3–4.6)	<10^−307^
	Total bilirubin (mg/dL)	0.7 (0.6–0.9)	0.6 (0.5–0.7)	<10^−307^
Hematological parameters	Leukocyte count (10^3^/μL)	5.9 (4.9–7.0)	5.6 (4.6–6.6)	0.2571
	Hematocrit (%)	44.9 (42.9–47.1)	40.0 (37.9–42.1)	<10^−307^
	Platelet count (10^3^/μL)	221.0 (190.0–256.0)	246.0 (211.0–286.0)	<10^−307^
	Red blood cell count (10^6^/μL)	5.1 (4.8–5.3)	4.5 (4.3–4.8)	<10^−307^
	Hemoglobin (g/dL)	15.1 (14.4–15.9)	13.2 (12.4–13.8)	<10^−307^

HbA1C: hemoglobin A1C, BP: blood pressure, LDL-C: low-density lipoprotein cholesterol, HDL-C: high-density lipoprotein cholesterol, AST: aspartate aminotransferase, ALT: alanine aminotransferase, γ-GT: γ-Glutamyl transferase, eGFR: estimated glomerular filtration rate, BUN: blood urea nitrogen. *p*: adjusted for age, BMI, and current smoking. Age: adjusted for BMI and current smoking. BMI: adjusted for age and smoking. * were analyzed with the exclusion of participants with previous history of hypertension; ** were analyzed with the exclusion of participants with previous history of diabetes mellitus; *** were analyzed with the exclusion of participants with previous history of gout; **** were analyzed with the exclusion of participants with previous history of hyperlipidemia. Data are presented as median (interquartile range).

**Table 2 ijms-22-10641-t002:** Association between *MUC1* rs12411216 and rs4072037 genotypes and clinical and laboratory parameters in Taiwan Biobank participants.

Clinical and Laboratory Parameters		rs12411216				rs4072037			
		Beta	SE	*p* Value	Adjusted *p* Value	Beta	SE	*p* Value	Adjusted *p* Value
Anthropology	Age (years)	−0.0604	0.0639	0.3444	0.9999	−0.0622	0.0637	0.3286	0.9999
	Waist circumference (cm)	−0.1125	0.0305	0.0002	0.0056	−0.1135	0.0304	0.0002	0.0056
	Waist–hip ratio	−0.0006	0.0003	0.0449	0.9999	−0.0006	0.0003	0.0484	0.9999
	Body mass index (kg/m^2^)	0.0294	0.0217	0.1764	0.9999	0.0285	0.0216	0.1873	0.9999
Blood pressure	Systolic BP * (mmHg)	0.2934	0.0918	0.0014	0.0392	0.3077	0.0914	0.0008	0.0224
	Diastolic BP * (mmHg)	0.1211	0.0598	0.0430	0.9999	0.1302	0.0596	0.0289	0.8082
	Mean BP * (mmHg)	0.1785	0.0658	0.0066	0.1848	0.1894	0.0655	0.0038	0.1064
Lipid profiles	Total cholesterol **** (mg/dL)	−0.0009	0.0005	0.0716	0.9999	−0.0008	0.0005	0.0796	0.9999
	HDL-cholesterol **** (mg/dL)	0.0003	0.0006	0.6180	0.9999	0.0004	0.0006	0.5337	0.9999
	LDL-cholesterol **** (mg/dL)	−0.0011	0.0007	0.1238	0.9999	−0.0011	0.0007	0.1234	0.9999
	Triglyceride **** (mg/dL)	−0.0025	0.0013	0.0548	0.9999	−0.0026	0.0013	0.0486	0.9999
Glucose metabolism	Fasting plasma glucose ** (mg/dL)	−0.2957	0.0903	0.0011	0.0308	−0.2788	0.0899	0.0019	0.0532
	HbA1C ** (%)	−0.0129	0.0036	0.0003	0.0084	−0.0125	0.0036	0.0005	0.0140
Uric acid	Uric acid *** (mg/dL)	0.0690	0.0067	5.73 × 10^−25^	1.60 × 10^−23^	0.0704	0.0067	4.02 × 10^−26^	1.13 × 10^−24^
Renal function	Creatinine (mg/dL)	0.0059	0.0013	4.00 × 10^−6^	0.0001	0.0061	0.0013	2.00 × 10^−6^	0.0001
	eGFR (mL/min/1.73 m^2^)	−1.0751	0.1294	9.86 × 10^−17^	2.76 × 10^−15^	−1.1074	0.1289	8.51 × 10^−18^	2.38 × 10^−16^
	BUN (mg/dL)	0.2540	0.0210	1.52 × 10^−33^	4.26 × 10^−32^	0.2558	0.0209	2.82 × 10^−34^	7.90 × 10^−33^
	Albuminuria (mg/L)	−0.0159	0.0027	5.31 × 10^−9^	1.49 × 10^−7^	−0.0158	0.0027	6.09 × 10^−9^	1.71 × 10^−7^
Liver function	AST (U/L)	0.1001	0.0727	0.1682	0.9999	0.1023	0.0723	0.1572	0.9999
	ALT (U/L)	−0.0186	0.1121	0.8681	0.9999	−0.0219	0.1116	0.8444	0.9999
	γ-GT (U/L)	0.3914	0.1870	0.0363	0.9999	0.4228	0.1862	0.0232	0.6496
	Serum albumin (g/dL)	0.0004	0.0013	0.7656	0.9999	0.0006	0.0013	0.6584	0.9999
	Total bilirubin (mg/dL)	−0.0014	0.0016	0.3672	0.9999	−0.0014	0.0016	0.3784	0.9999
Hematological parameters	Leukocyte count (10^3^/μL)	0.0044	0.0092	0.6307	0.9999	0.0042	0.0092	0.6494	0.9999
	Hematocrit (%)	−0.1509	0.0209	5.07 × 10^−13^	1.42 × 10^−11^	−0.1497	0.0208	6.34 × 10^−13^	1.78 × 10^−11^
	Platelet count (10^3^/μL)	−0.7588	0.3417	0.0264	0.7392	−0.7417	0.3402	0.0293	0.8204
	Red blood cell count (10^6^/μL)	−0.0153	0.0026	7.49 × 10^−9^	2.10 × 10^−7^	−0.0147	0.0026	2.22 × 10^−8^	6.22 × 10^−7^
	Hemoglobin (g/dL)	−0.0540	0.0074	2.42 × 10^−13^	6.78 × 10^−12^	−0.0534	0.0073	3.48 × 10^−13^	9.74 × 10^−12^

Abbreviations, adjusted conditions, and subjects recruited for analysis as in Table 1. Adjusted *p* value: with Bonferroni correction, *n* = 28. * were analyzed with the exclusion of participants with previous history of hypertension; ** were analyzed with the exclusion of participants with previous history of diabetes mellitus; *** were analyzed with the exclusion of participants with previous history of gout; **** were analyzed with the exclusion of participants with previous history of hyperlipidemia.

**Table 3 ijms-22-10641-t003:** Lead single-nucleotide polymorphisms at the *TRIM46–MUC1–THBS3–MTX1* gene region.

Phenotype	*p* Value	Lead SNP	Gene Locus	Position	Allele	MAF	Location	SNP Function/Amino Acid (Codon)
Waist circumference	0.0001	rs760077	*MTX1*	155178782	T/A	0.1869	Missense variant	p.Ser63Thr
HbA1C	0.0002	rs914615	*THBS3*	155175892	A/G	0.1895	Intron variant	TFBS
Uric acid level	3.63 × 10^−33^	rs2070803	*TRIM46–MUC1*	155157715	G/A	0.2350	3′ UTR	--
Serum creatinine	0.0004	rs76872124	*TRIM46*	155152205	C/T	0.1119	Synonymous variant	p. His438His
eGFR	3.06 × 10^−19^	rs2070803	*TRIM46–MUC1*	155157715	G/A	0.2350	3′ UTR	--
Serum urea nitrogen	2.78 × 10^−35^	rs4971101	*TRIM46–MUC1*	155157635	A/G	0.2320	3′ UTR	--
Albuminuria	2.21 × 10^−10^	rs423144	*THBS3*	155169355	T/G	0.2309	Intron variant	--
Hematocrit	2.1 × 10^−15^	rs4971093	*KRTCAP2–TRIM46*	155144300	A/G	0.2193	2 Kb upstream variant (*TRIM46*)	TFBS
Red blood cell count	1.18 × 10^−9^	rs76872124	*TRIM46*	155152205	C/T	0.1119	Synonymous variant	p. His438His
Hemoglobin	2.27 × 10^−14^	rs76872124	*TRIM46*	155152205	C/T	0.1119	Synonymous variant	p. His438His
Gout	2.94 × 10^−8^	rs12411216	*MUC1*	155164480	A/C	0.2053	2 Kb upstream variant	TFBS
Microalbuminuria	0.0031	rs4971101	*TRIM46–MUC1*	155157635	A/G	0.2320	3′ UTR	--

Abbreviations as in Table 1. TFBS: transcription factor binding site, SNP: single-nucleotide polymorphism, MAF: minor allele frequency, UTR: un-translated region.

## Data Availability

The data presented in this study are available on request from the corresponding author.

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
