# Peer review of "Pleiotropic Effects of Functional MUC1 Variants on Cardiometabolic, Renal, and Hematological Traits in the Taiwanese Population"

_ijms, 2021, doi:10.3390/ijms221910641_

Round 1
Reviewer 1 Report
The authors describe their work on the pleitropic effects of functional MUC1 variants on cardiometabolic, renal, and hematological traits in the Taiwanese people. It was found that the pleitropic effects of MUC1 variants with novel associations with gout, red blood cell parameters and MUC1 DNA methylation. It was concluded that TRIM46-MUC1-THBS3-MTX1 gene region variantsmay help in further understanding of the pathogenesis of cardiometabolic, renal, and hematological disorders. This is an interesting study. Appropriate methodology has been employed and the conclusions appear to be justified based on the data at hand. I have a few minor points for consideration.
- Please provide a clear hypothesis to be tested in the study.
- Can the authors increase the size of Figs 3 and 4 as it is difficult to read.
- The authors should elaborate and emphasize the novelty aspect of their work as well as expand on the clinical applicability of their findings.
- Are there any comparable studies conducted in other genetically distinct populations, or is this a genetic trait unique to Taiwanese people?
- What percent of the disease (cardiometabolic, renal disease, cancer etc) are associated with MUC1 variants?
Author Response
Q1: Please provide a clear hypothesis to be tested in the study.
Ans: Thank you for your comments. We have added a hypothesis to be tested in the study by adding in the Introduction section of the revised manuscript with “We hypothesized that both genetic and epigenetic effects of MUC1 gene may play a crucial role in the pathogenesis of cardiometabolic, renal, and hematological disorders.”.
Q2: Can the authors increase the size of Figs 3 and 4 as it is difficult to read.
Ans: Thank you for your comments. We have increased the size of Figs 3 and 4, as suggested by the reviewer.
Q3: The authors should elaborate and emphasize the novelty aspect of their work as well as expand on the clinical applicability of their findings.
Ans: Thank you for your comments. We have added in the first part of the Discussion section by adding in the revised manuscript: “We also provide the first evidence of the possibility that the previously reported association between rs4072037 and the total or alternative spliced form of gene expression may be secondary to MUC1 gene-body methylation status.”.
Q4: Are there any comparable studies conducted in other genetically distinct populations, or is this a genetic trait unique to Taiwanese people?
Ans: Thank you for your comments. Both MUC1 variants, rs4072037 and rs12411216, are common variants in different ethnic populations with only some differences in the minor allele frequencies. It is likely that the results in the manuscript are not unique to Taiwanese people but can be applied to other populations as shown in many references in the manuscript.
Q5: What percent of the disease (cardiometabolic, renal disease, cancer etc) are associated with MUC1 variants?
Ans: Thank you for your comments. Cardiometabolic, renal, and cancer disorders are all multifactorial disorders and are affected by various genetic, environmental, and genetic-environmental interactions. It is difficult to estimate the percentage of the disease that are affected by MUC1 variants, however, the genome-wide association in these functional variants with multiple metabolic, biochemical, and hematological parameters suggesting the critical role of TRIM46–MUC1–THBS3–MTX1 gene region variants on these disorders.
Reviewer 2 Report
Ming-Sheng et al. have investigated the pleiotropic effects of different functional variants, mainly rs4072037 and rs12411216, on cardiometabolic, renal and hematological traits in the Taiwanese populations. More than 80,000 participants were enrolled in this analytical study and they ran different biostatistical analysis. They concluded that above mentioned variants of MUC1, influences the waist circumference, systolic blood pressure, HbA1C, renal functional parameters, albuminuria, RBC counts. It was also associated with high serum uric acid levels and gout risk. These variants were also associated with MUC1 gene methylation. Regional-plot analysis also revealed that the phenotype is associated with TRIM46-MUC1-THBS3-MTX1 gene region on chromosome 1q22.
The manuscript is well written and claims made by the authors are well supported by the data presented in this manuscript. Thus, this manuscript can be accepted in its current form.
Author Response
Thanks for the comments of the reviewer.